# Selection of Optimal Operating Conditions for Extraction of *Myrtus Communis* L. Essential Oil by the Steam Distillation Method

**DOI:** 10.3390/molecules25102399

**Published:** 2020-05-21

**Authors:** Durmuş Alpaslan Kaya, Mihaela Violeta Ghica, Elena Dănilă, Şevket Öztürk, Musa Türkmen, Mădălina Georgiana Albu Kaya, Cristina-Elena Dinu-Pîrvu

**Affiliations:** 1Department of Field Crops, Faculty of Agriculture, Mustafa Kemal University, 31034 Antakya-Hatay, Turkey; dak1976@msn.com (D.A.K.); turkmenmusa@hotmail.com (M.T.); 2Department of Physical and Colloidal Chemistry, Faculty of Pharmacy, University of Medicine and Pharmacy “Carol Davila”, 20956 Bucharest, Romania; ecristinaparvu@yahoo.com; 3Faculty of Applied Chemistry and Materials Science, University Politehnica of Bucharest, 011061 Bucharest, Romania; 4Department of Collagen, Division Leather and Footwear Research Institute, National Research and Development Institute for Textile and Leather, 031215 Bucharest, Romania; albu_mada@yahoo.com

**Keywords:** *Myrtus communis* L. essential oil, factorial design, Taguchi approach, GC-MS assay

## Abstract

*Myrtus communis* L. is one of the important aromatic and medicinal species from the Mediterranean area. It is used in various fields such as culinary, cosmetic, pharmaceutical, therapeutic, and industrial applications. Thus, a Box–Wilson experimental plan was used in this study to select the optimal operating conditions in order to obtain high volumes of essential oils. The factorial design method was applied to evaluate at an industrial scale the effect of major process variables on the essential oil extraction from *Myrtus communis* L. herbs by the steam distillation method. The input variables considered as significant operating conditions were: X_1_—boiler occupancy rate (boilers were filled to 50%, 75%, and 100%), X_2_—distillation duration (distillation was continued 60, 75, and 90 min), and X_3_—particle size (herbs were cut in sizes of 10, 20, and 30 mm via guillotine). The dependent variable selected, coded as Y, was the essential oil volume obtained (mL). The steps of the classical statistical experimental design technique were complemented with the Taguchi method to improve the extraction efficacy of essential oil from *Myrtus communis* L., and the optimum parameter conditions were selected: boiler occupancy rate 100%, distillation duration 75 min, and particle size 20 mm. Following the optimum parameters, the GC-MS assay revealed for the *Myrtus communis* L. essential oil two predominant components, α-pinene—33.14% and eucalyptol—55.09%.

## 1. Introduction

*Myrtus communis* L., known as Myrtle, is one of the important aromatic and medicinal species from the Mediterranean area [1]. It is used in various fields such as culinary, cosmetic, pharmaceutical, therapeutic, and industrial applications [2,3,4]. Different species of myrtle showed the presence of essential oils, phenolic acids, flavonoids (quercetin, catechin, myricetin), tannins, anthocyanin, pigments, and fatty acids [3,5,6]. Essential oils of *Myrtus* species consist mainly of monoterpene hydrocarbons, oxygenated monoterpenes, ethers, esters, sesquiterpene hydrocarbons, oxygenated sesquiterpenes, aliphatic hydrocarbons, alcohols, and phenols distributed in various ratios depending on the geographical area (temperature, soil quality, day length), harvesting time, and genotype of the species [7,8,9,10].

*Myrtus communis* is an ancient sacred plant which was used in ceremonies and as a symbol of youth and beauty. In ancient mythology, myrtle was sacred to Aphrodite and become the plant of love. The name derives from the Greek Myron meaning balm, chrism, ointment. It has been known to Persians, Greeks, and other civilized nations since ancient times and is native to South America, Australia, the north-western Himalayas, the Middle East, West Asia, North Africa, and the Zagros mountain range regions but was first introduced to the world by the British in 1597. *Myrtus communis* grows in Mediterranean countries along the coastline and inland hills in spruce and pine forests and near riversides and it is also known as Myrtle, Hambeles, Mersin, and Murt [11]. 

Myrtle is an evergreen tree that grows to a height of about 1–5 m, with small leaves ovate or lanceolate, 2–5 cm long, and white flowers. This species is a very aromatic plant because of the high essential oil content in its leaf, flower, and fruit glands; the leaves also contain tannins and flavonoids [12,13].

Antimicrobial (antibacterial, antifungal, and antiviral) and antioxidant proprieties of compounds produced by *Myrtus communis* L. have been reported in numerous studies. For example, 1,8-cineol and linalool have shown a good bactericidal effect against Gram positive and Gram negative bacteria (*Listeria monocytogenes*, *Streptococcus pyogenes*, *Proteus vulgaris,* and *Escherichia coli*) [14]. Another study showed that some myrtle single compounds such as limonene, 1,8-cineole, and α-pinene have significant activity against *Mycobacterium tuberculosis* [1].

The myrtle essential oil proved to have anti-inflammatory, anticancer, antiviral, antimicrobial, antifungal, antioxidant, antidiabetic, antiulcer, and anthelmintic properties [15,16,17,18,19].

Taking into account these therapeutic properties of myrtle essential oil, one of the top subjects in the essential oil industry is to obtain both high extracted volumes and optimal conditions of extraction with reduction of experimental trials [20]. The recent studies showed the use of response surface methodology (RSM) as effective and powerful statistical method to optimize the essential oils extraction. For example, RSM was applied to determine at the laboratory scale the optimum extraction parameters to obtain high volumes of essential oil from *Citrus latifolia* by distilling peels [21], and high yields of essential oil from *Cyperus rotundus* Linn. rhizomes [22], *Piper betle* L. fresh and cured leaves [23], and *Pistacia lentiscus* ripe berries [24]. Moreover, Taguchi orthogonal array design was used to optimize the essential oil extraction yield from aerial parts of *Dracocephalum kotschyi Boiss* [25]. The essential oils extraction process from *Cymbopogon martinii* (Palmarosa) leaves was studied by the Taguchi method combined with grey relational analysis [26]. The same Taguchi experimental design was applied to extract biodiesel from *Manilkara zapota* L. seed oil [27].

Also, the statistical optimization techniques were applied to analyze and optimize the extraction at laboratory scale of essential oil from *Myrtus communis* L. leaves originating from different countries, using various techniques. 

Zermane et al., found that the supercritical fluid extraction (SCCO_2_) process used to obtain essential oils from the Algerian myrtle leaves resulted in the maximum yield of 4.89 wt% for a temperature of 313 K, pressure of 30 Mpa, and the lowest mean particle diameter [28]. Perreira et al., showed that the optimum experimental conditions resulting in a high yield of essential oil from Portugal myrtle leaves, using the same supercritical CO_2_ extraction method, refers to a pressure of 22 MPa, temperature of 51 °C, and SCCO_2_ flow rate of 0.1 kg/h [29]. Haj Ammar et al., studied the influence of three technological parameters involved in the hydrodistillation process on essential oil yield extracted from Tunisian myrtle leaves, and found that the ratio of vegetal material weight/water weight and the condensation flow had the most significant effect while the ultrasound exposure time effect can be neglected [30]. In another study by Haj Ammar et al., a wider screening design was used to find the best conditions for a higher extraction yield also from Tunisia myrtle leaves. In this case, the results indicated that the yield is mostly influenced by processing time, condensation flow, and the mass ratio of plant material:water, while the granulometry had a less important effect [31]. Ghasemi et al., performed a more complex study, analyzing the results for the essential oil yield obtained from Iranian myrtle leaves by two extraction techniques: supercritical fluid extraction using different operating conditions and hydrodistillation methods, concluding that the first method led to the best results [32].

Thus, the aim of this study was to select the optimal operating conditions in order to obtain high volumes of essential oils using a Box–Wilson factorial design complemented with response surface analysis and the Taguchi approach. This optimization method was successfully applied by the authors in the field of drug delivery systems to get the best values for the physical–chemical and biopharmaceutical parameters, as well as the most stable, robust, and insensitive to the noise factor responses [33,34,35,36]. To the best of our knowledge, in this study we investigated for the first time at an industrial scale the effect of major process variables (boiler occupancy rate, distillation time, and size of particles) on the volume of essential oil extracted from *Myrthus communis* L. herbs by steam distillation. 

## 2. Results and Discussion

### Design of Experiments and Optimization Technique

A 3-factor, 3-level Box–Wilson experimental plan was applied to establish the best extraction conditions of essential oils from *Myrtus communis* L. herbs at an industrial scale.

While a full 3^3^ factorial design requires 27 experiments, the Box–Wilson experimental plan reduces the number of trials to 15. The Box–Wilson factorial design is also a face centered composite design that includes eight factorial points corresponding to 2^3^ full factorial design, six axial points corresponding to the face centers of the cube portion of the design, and one replicate at the centre of the design [33].

The essential oil volumes (mL) obtained during the extraction process from *Myrtus communis* L. herbs and the Box–Wilson experimental plan used to conduct the experiments are summarized in Table 1. 

The experimental data from the Box–Wilson design were subjected to the optimization technique based on the experimental design and response surface methodology complemented with Taguchi approach elements [33,34,35,36]. In the first stage of the optimization process, a stepwise regression analysis with a backward elimination subroutine was applied to the experimental data for setting out the reduced quadratic polynomial equation for the response (Equation (1)).
(1)$$Y=18.518{\;X}_{3}-0.0543{\;X}_{2}^{2}-0.464{\;X}_{3}^{2}+0.122{\;X}_{1}X_{2}$$

In Equation (1) only the significant terms (*p* < 0.05) were considered and indicate the interaction and quadratic effects of the extraction process operating conditions on the essential oil volume (Table 1). The regressional coefficient values in the above equation explain their influence on the selected response. For a response that has to be maximized according to the constraint from Table 6, a positive sign means a synergistic effect while a negative sign means an antagonistic effect of the corresponding input variables. Thus, the coefficients of the reduced model in Equation (1) show that for X_3_ a positive linear effect is noticed, while for both quadratic forms of X_2_ and X_3_ a negative effect appears. The essential oil volume is positively influenced by the interaction between X_1_ and X_2_.

The above reduced quadratic polynomial equation was assessed through the determination coefficient (R^2^), correlation coefficient (R), analysis of variance (ANOVA), and residual analysis, respectively. The R^2^ value of 0.9948 was higher than 0.90, while R value was 0.9974, closed to 1. ANOVA results proved the statistical significance of the regressional model, the value for the model probability being smaller than <0.0001. The results of the residual analysis are listed in Table 1. All these results indicated a good predictive power of the reduced regressional equation. The summary of variance analysis is presented in Table 2, indicating the statistical significance of the regression model.

The observed and predicted values are well correlated as resulting from the residual analysis given in Table 1 and Figure 1a, in which a linear distribution is noticed.

Also, the difference between observed and predicted responses expressed as normal probability plots of residuals validated the design normality. The robustness of the design is confirmed by the distribution near a straight line of the response experimental values (Figure 1b).

The relationship between the dependent variables and two independent variables were further analyzed by response surface methodology. Three-dimensional (3D) response surface graphs were built (Figure 2a–c). The 3D graphs allow the visualization of the combined effects of the operating parameters on the essential oil volume obtained under different experimental conditions.

From Figure 2a,b it is determined that the best values for the essential oil volume are obtained for a higher boiler occupancy rate. Thus, an increase of 171.10% for the essential oil volume (from 275 mL to 745 mL) was recorded when the boiler occupancy rate varied from minimum to maximum levels and the duration of distillation was kept at the minimum level. For the duration of the distillation increase from the minimum to the maximum level when the boiler occupancy rate was at the maximum level, the essential oil volume increased only 4.11% (from 730 mL to 760 mL). A similar dependency was recorded for the influence of the boiler occupancy rate and the particle size on the essential oil volume (Figure 1b).

The essential oil volume as a function of the duration of distillation and particle size (Figure 2c) indicates that its highest values are obtained for middle particles sizes, and medium to high duration of distillation.

Taking into consideration all remarks above, we could conclude that the optimum variation levels for operating parameters in the extraction process of the essential oil from *Myrtus communis* L. herbs are as follows: X_1_: [70 ÷ 100] %, X_2_: [70 ÷ 90] min, X_3_: [10 ÷ 20] mm. It seems that a higher boiler occupancy rate and duration of distillation conduct an increase of the essential oil volume, while the same effect is recorded for smaller particle size of the leaves.

The steps of the statistical experimental design technique previously detailed were complemented with the Taguchi method to improve the quality of the extraction process of essential oil from *Myrtus communis* L. herbs. The noise impact on the characteristic target was mathematically evaluated through the signal-to-noise ratio (S/N). 

The signal-to-noise ratio for a criterion that has to be maximized coded as “larger-the-better” was computed (Table 3) for each experiment from Box–Wilson design, being specific to the dispersion around the value of the analyzed response for the combination of the tested operating parameters for the extraction process.

The control factors (independent variables X_1_–X_3_) effects on the S/N for the selected response (Y), resulting in the optimal combination of the extractive operational conditions, are given in Table 4 and Figure 3.

From Figure 3 it can be noticed that X_1_ parameter has the most significant influence on the Y response. The optimal coded level of this formulation factor was 3, meaning that this level involved a reduction of the noise factors effect.

For X_2_ and X_3_ a smaller influence on the response was remarked, in both cases the noise factors effect reduction was consequently obtained for the coded level 2.

The effect size (Table 4) of the operating conditions on the S/N ratio gives the information concerning their influence degree on the system responses. Thus, the boiler occupancy rate was the main influencing factor for the essential oil volume, the effect size being 8.36 times higher than the duration of distillation and 6.72 times higher than the particle size.

Using the Taguchi technique, a combination of the operating parameters belonging to the initial fractional factorial design was selected (*Run 10*). The response obtained did not have the biggest value (see Table 1) but was the most robust, stable, and insensitive to the noise factors. For this experiment, close values are obtained for the S/N ratio between the predictive value (58.42 dB) and the experimental one (57.50 dB).

The essential oil obtained in high volumes was characterized by GC-MS and the components were obtained as shown in Table 5.

The GC-MS assay revealed for this essential oil obtained at industrial scale two components in high amounts: α-pinene—33.14% and eucalyptol—55.09%. Other compounds found in amounts of more than 1% were: β-fenchyl alcohol (3.20%), α-terpinenyl acetate (1.93%), linalool (1.79%) and 3(10)-caren-4-ol, and acetoacetic acid ester (1.08%). The remaining identified components amount was only 3.77%. 

In the chemical composition some differences compared to literature studies for essential oil extracted by different procedures at laboratory scale were noticed. Some examples of chemical composition of the myrtle essential oil from leaf belonging to different regions and described by many authors are given hereafter.

Ghannadi and Dezfuly identified the major constituents of *M. communis* leaf essential oil from Iran, extracted through hydrodistillation method, as: α-pinene (37.8%), 1,8-cineole (23.1%), limonene (17.1%), and linalool (10.1%) [37].

Walle et al., identified myrtenol, myrtenol acetate, limonene (23%), linalool (20%), α-pinene (14%), and 1,8-cineol (11%) as the most important constituents of myrtle essential oil from Ethiopia, obtained also by the hydrodistillation method [38]. 

According to Senatore et al., and using the same classical extraction method, linalool (36.5%) and linalyl acetate (16.3%) were found as the most abundant component in the Turkish myrtle essential oil, with other compounds in representative amount being 1,8-cineole (10.5%), geranyl acetate (8.0%), α-terpineol (7.9%), and α-pinene (7.8%) [39].

The oil extracted by hydrodistillation from leaves of plants grown in various locations in Sardinia contained mostly α-pinene (generally 30.0%, but reaching in one case a maximum of 59.5%), 1,8-cineole (ranging from 15.9 to 41.7% with an average of 28.8%), and limonene (ranging from 5.2 to 29.8% with an average of 17.5%) as the most abundant components [40].

Spanish oil obtained using the hydrodistillation technique was characterized by a high content of myrtenyl acetate (35.9%), 1,8-cineole (29.89%), and a low content of α-pinene (8.2%) and limonene (7.58%) [41]. 

The oil from Greece, extracted by steam distillation, was found to be rich in linalyl acetate (31.4%), limonene (21.8%), α-pinene (18%), and geranyl acetate (6.5%) [42].

According to Ghasemi et al., differences between the number and amount of compounds identified for Iranian essential oil extracted by hydrodistillation and supercritical carbon dioxide were recorded [32]. The selectivity of the extraction was obvious for the SCCO_2_ where three major components were predominant (α-pinene (29.9 to 38.6%), limonene (13.5 to 18.1%), and 1,8 cineole (23.3 to 29.1%)) in comparison with the conventional method (α-pinene (31.8%), 1,8-cineole (24.6%), limonene (14.8%), and linalol (8.3%)).

Tunisian oil obtained by steam distillation showed major components like myrtenyl acetate (20.75%), 1,8-cineol (16.55%), α-pinene (15.59%), linalool (13.30%), and limonene (8.94%) [8].

This composition variations are generated by a series of factors, such as: pedo-climatic conditions, harvesting period, different subspecies, extraction method applied, and operating conditions.

## 3. Materials and Methods

### 3.1. Materials

The *Myrthus communis* L. herbs were harvested during the full bloom period, when the amount of active substance was most intense, being dried after in the shade at room temperature.

### 3.2. Methods

#### 3.2.1. Obtaining Essential Oils

Essential oils were obtained by the steam distillation extraction method from *Myrthus communis* L. herbs using special equipment developed by Mahan Cosmetics in Hatay, Turkey, and varying the following parameters: boiler occupancy rate, distillation duration, and particle size. 

The distillation equipment consisted of 4 units, 2 active and 2 passive. When the active units were distilled, passive units were ready for distillation by discharging, cleaning, and filling. When the duration of the distillation of the active units reached the end, they were replaced by passive units and thus 24/7 continuous distillation was possible. Each distillation unit was 3 m high and had 500 L of water capacity. Water used in distillation was free of lime and heavy metals. With the automation program used, the parameters such as pressure, temperature, and time, which are exposed to the oiled plant during the distillation, were recorded and could be traced backwards. Low temperature (98 °C) and water vapor pressure (1 bar) were used to keep the quality and yield of volatile oil high. Furthermore, the water vapor (101 °C) was directed through the gap between the two walls to prevent the inner wall from being cooled, evaporated, and condensed repeatedly and consequently to prevent deterioration of essential oil quality.

#### 3.2.2. Design of Experiments and Optimization Techniques

In order to establish the best extraction conditions of essential oils from *Myrtus communis* L. herbs by the steam distillation method, a Box–Wilson factorial design was used (Table 6). 

A 3-factor, 3-level Box–Wilson experimental plan was applied to establish the best extraction conditions of essential oils from *Myrtus communis* L. herbs at an industrial scale.

The input variables considered as significant operating conditions were boiler occupancy rate, distillation duration, and particle size, coded as X_1_, X_2_, and X_3_. The boiler occupancy rate was varied between 50% and 100%, the distillation duration was continued between 60 and 90 min, and the herbs were cut to particle sizes between 10 and 30 mm via guillotine. Each independent variable was evaluated at three different coded levels: low, middle, and high, coded as 1, 2, and 3, respectively, and their values (under the uncoded and coded form) are given in Table 6. The dependent variable as the system response selected in the Box–Wilson design was the essential oil volume obtained (mL) coded as Y, its constraint being listed in Table 6.

The experiments were randomly performed. The following response function approximated by the second degree polynomial equation (Equation (2)) was used to correlate the dependent variable (Y) with input parameters (X_1_–X_3_):(2)$$Y=\beta_{0}+\beta_{1}X_{1}+\beta_{2}X_{2}+\beta_{3}X_{3\;}+{\;\beta }_{11}X_{1}^{2}+{\;\beta }_{22}X_{2\;}^{2}+{\;\beta }_{33}X_{3}^{2}+{\;\beta }_{12}X_{1}X_{2}+{\;\beta }_{13}X_{1}X_{3}+{\;\beta }_{23}X_{2}X_{3}$$
where $\beta_{0}$ is the model constant, $\beta_{1}–\beta_{3}$ the linear coefficients,${\;\beta }_{11}–\beta_{33}$ the quadratic coefficients, and $\beta_{12},\;\beta_{13},\;\beta_{23}$ the cross-products coefficients. Statistica StatSoft Release software package was used for the determination of the coefficients of Equation (1) by regressional analysis of the experimental data. A stepwise regression analysis was conducted to build the second order polynomial equations for the response variable. The significant terms (*p* < 0.05) were selected for the final equation and the reduced quadratic polynomial model was obtained. The best fitting mathematical model was selected based on the determination coefficient, correlation coefficient, analysis of variance, and residual analysis. To investigate the combined effect of independent variables on the response, three-dimensional response surfaces were also drawn. The Taguchi signal/noise ratio was finally used for the evaluation of the design robustness. 

##### Taguchi Technique

Taguchi’s technique is one of the most known methods for a robust experimental design that ensures the optimization of the product and the conditions of the process to obtain it. The robustness represents the quality of being able to overpass the on-going modifications. The Taguchi technique is a tool for process improvement and not for absolute optimization. In the frame of Taguchi’s approach, the independent variables (X_1_–X_3_) are considered control factors, which need to be optimized for reaching a specified value and to eliminate the variation. Besides the control factors, the system responses can be affected by the noise factors, influenced by the process deployment conditions, which are defined as the unwanted variability determining the decrease of the optimization process quality. In order to find a robust solution, Taguchi developed more than 70 such signal-to-noise ratios (S/N), according to the particular type of the characteristics involved. Among these, three performance indicators are the most used, “lower-the-better”, “larger-the-better”, and “nominal-the-better”. In this case, taking into account the constraints imposed on dependent variable Y we used “larger-the-better” criterion, known in literature also as “more-is-better”or “higher-is-better”. This S/N ratio can be seen as a criterion that has to be minimized if the reversed measured data are taken into account, using the following equation (Equation (3)) [30,31]:(3)$$\frac{S}{N}=-10\log\;\left(\frac{1}{n}\overset{n}{\underset{i=1}{\sum }}\frac{1}{Y_{i}^{2}}\right)$$

#### 3.2.3. Gas Chromatography–Mass Spectrometry (GC-MS) Analysis

Essential oils from myrtle plants were stored in amber vial bottles at +4 °C until analysis by GC–MS (Thermo Fisher Scientific, Milan, Italy). A sample of 5 µL of essential oil taken from the essential oils stored for GC–MS analysis with the help of a micro syringe was injected into 2 mL vials containing cyclohexane. Analysis of the essential oils obtained in high volume was carried out by using a Thermo Scientific DSQ model Gas Chromatograph equipped with MS, auto sampler, and TR-5MS (5% phenyl polysilphenylenesiloxane, 0.325 mm × 60 m i.d, film thickness 0.25 µm). The carrier gas was helium (99.9%) at a flow rate of 1 mL/min; ionization energy was 70 eV. Mass range was *m*/*z* 50–650 amu. Data acquisition was in scan mode. MS transfer line temperature was 280 °C, MS ionization source temperature was 200 °C, the injection port temperature was 250 °C. The samples were injected with a 250 split ratio. The injection volume was 1 μL. Oven temperature was programmed in the range of 50 to 220 °C at 3 °C/min. The structure of each compound was identified by comparison with their mass spectrum (Wiley9 library). The data were handled using the Xcalibur software program (version 2.1.0 SP1, Thermo Fisher Scientific, Milan, Italy).

## 4. Conclusions

In order to obtain high volumes of myrtle essential oils a Box–Wilson factorial design was applied to evaluate at an industrial scale the effect of major process variables on the essential oil extraction from *Myrtus communis* L. herbs by the steam distillation method. The steps of the classical statistical experimental design technique were complemented with the Taguchi method to improve the quality of the extraction process of essential oil from *Myrtus communis* L., and the optimum parameter conditions were selected as follows: boiler occupancy rate 100%, distillation duration 75 min, and particle size 20 mm. Following the optimum parameters, the GC-MS assay revealed for the *Myrtus communis* L. essential oil two predominant components, α-pinene—33.14% and eucalyptol—55.09%.

## Figures and Tables

**Figure 1 molecules-25-02399-f001:**
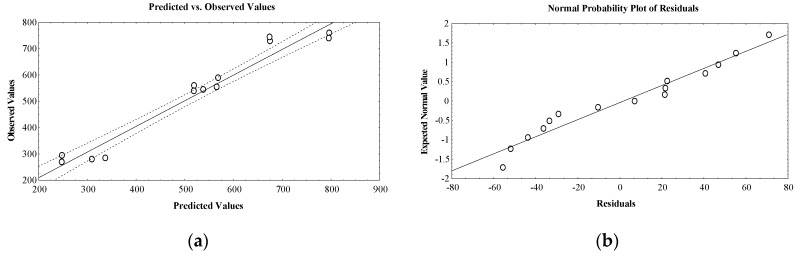
(**a**) Plot showing correlation between observed and predicted values for essential oil volume (mL); (**b**) plot showing correlation between expected normal values and residuals for essential oil volume (mL).

**Figure 2 molecules-25-02399-f002:**
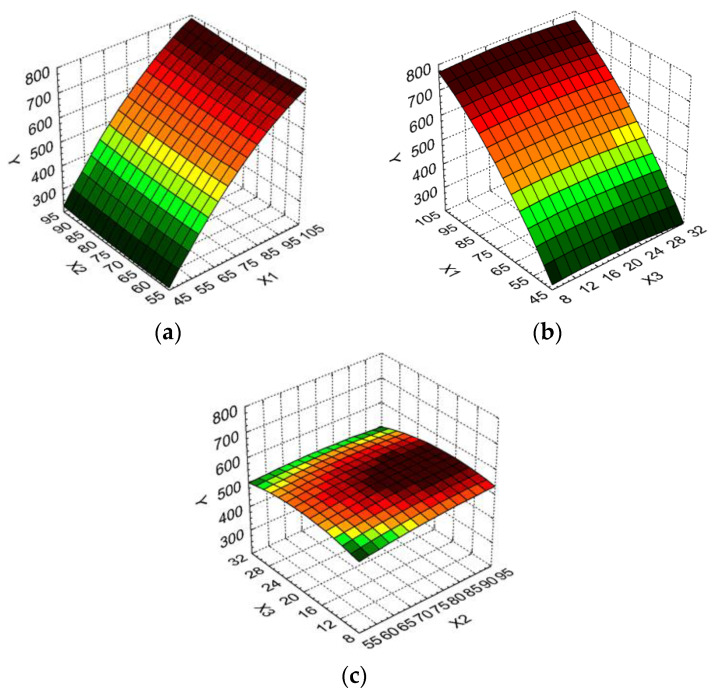
3D response surface and contour plot showing the effect of different operating conditions on essential oil volume (Y): (**a**) boiler occupancy rate (X_1_) and duration of distillation (X_2_); (**b**) boiler occupancy rate (X_1_) and particle size (X_3_); (**c**) duration of distillation (X_2_) and particle size (X_3_).

**Figure 3 molecules-25-02399-f003:**
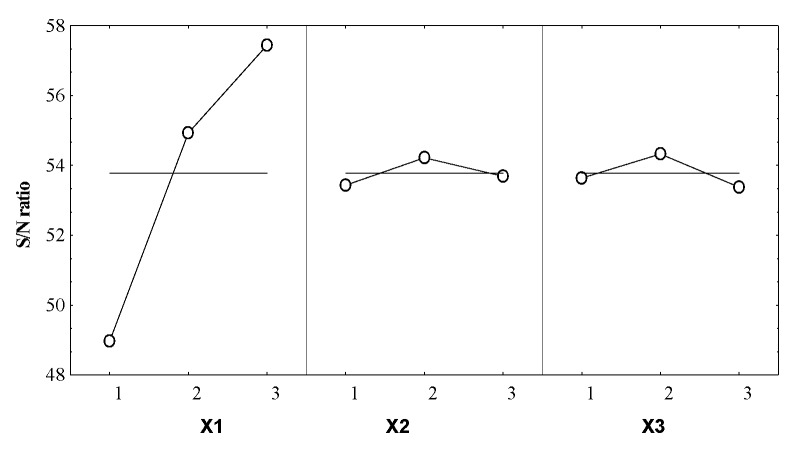
Control factors effects on the S/N ratio for the essential oil volume.

**Table 1 molecules-25-02399-t001:** Coded values and physical levels of the variables used in different experimental trials of the fractional matrix Box–Wilson and the corresponding observed and predictive responses.

Trials No.	Input Variables Coded Level (Physical Level)			Response	
	X_1_ (%)	X_2_ (min)	X_3_ (mm)	Y (mL)	
				Obs.	Pre.
1	1 (50)	1 (60)	1 (10)	280	309
2	3 (100)	1 (60)	1 (10)	730	675
3	1 (50)	3 (90)	1 (10)	295	248
4	3 (100)	3 (90)	1 (10)	760	796
5	1 (50)	1 (60)	3 (30)	275	308
6	3 (100)	1 (60)	3 (30)	745	674
7	1 (50)	3 (90)	3 (30)	270	247
8	3 (100)	3 (90)	3 (30)	740	795
9	1 (50)	2 (75)	2 (20)	285	336
10	3 (100)	2 (75)	2 (20)	750	793
11	2 (75)	1 (60)	2 (20)	545	538
12	2 (75)	3 (90)	2 (20)	590	568
13	2 (75)	2 (75)	1 (10)	560	519
14	2 (75)	2 (75)	3 (30)	540	518
15	2 (75)	2 (75)	2 (20)	555	565

**Table 2 molecules-25-02399-t002:** Analysis of variance for the reduced regressional polynomial model.

Responses	Sources of Variation	Sum of Squares	df	Mean Squares	*F*-Value	*p*-Value
Y	Regression	4704598	4	1176150 2232	526.94	<0.0001
	Residual	24552	11			
	Total	4729150	15			

**Table 3 molecules-25-02399-t003:** The values for the signal/noise (S/N) ratio of the system response for the experiments included in the fractional factorial design.

Run Order	1	2	3	4	5	6	7	8	9	10	11	12	13	14	15
S/N (dB)	48.94	57.26	49.36	57.61	48.78	57.44	48.62	57.38	49.0	57.50	54.72	55.41	54.96	54.64	54.88

**Table 4 molecules-25-02399-t004:** Optimal combinations of independent variables coded levels, identified by the Taguchi technique, and their effect size on S/N ratio for the dependent variable, expected, and observed S/N value.

Control Factors (Independent Variables)	Y	
	“Larger—the—Better”	Effect Size
X_1_	3	3.662
X_2_	2	0.438
X_3_	2	0.545
S/N expected (dB)	58.42	
S/N observed (dB)	57.50	

**Table 5 molecules-25-02399-t005:** Essential oil components of *Myrtus communis* L. herbs.

RT	Compound Name	SI	RSI	CAS Number	%
12.99	α-Phellandrene	931	970	1529-99-3	0.24
13.38	α-Pinene	985	987	80-56-8	33.14
14.19	cis-Ocimene	681	786	6874-10-8	0.06
15.49	Δ-3-Carene	976	976	13466-78-9	0.43
15.96	β-Pinene	674	810	127-91-3	0.09
16.82	γ-Terpinene	958	978	99-85-4	0.29
18.08	Eucalyptol	984	985	470-82-6	55.09
20.73	α-Terpineolene	875	923	586-62-9	0.17
21.50	Linalool	978	980	78-70-6	1.79
24.44	trans-Pinocarveol	897	956	547-61-5	0.13
26.43	α-Terpineol	782	936	10482-56-1	0.06
27.00	Terpinen-4-ol	907	966	562-74-3	0.15
28.00	β-Fenchyl alcohol	938	955	470-08-6	3.20
33.73	trans-Pinocarvyl acetate	781	931	1686-15-3	0.12
35.27	3(10)-Caren-4-ol, acetoacetic acid ester	862	862	NA	1.08
36.50	α-Terpinenyl acetate	958	988	80-26-2	1.93
38.47	β-Elemene	759	891	515-13-9	0.04
39.15	Linalyl acetate	747	876	115-95-7	0.14
40.14	trans-Caryophyllene	955	970	87-44-5	0.19
41.96	α-Humulene	872	940	6753-98-6	0.08
48.07	Caryophyllene oxide	738	893	1139-30-6	0.06

**Table 6 molecules-25-02399-t006:** Process variables and experimental conditions in 3-factor, 3-level Box–Wilson experimental designs.

**Input Variables**	**Coded Symbol**	**Coded and Uncoded Variation Levels**		
		**Low (1)**	**Middle (2)**	**High (3)**
Boiler occupancy rate, (%)	X_1_	50	75	100
Duration of distillation, (min)	X_2_	60	75	90
Particle size, (mm)	X_3_	10	20	30
**Response**	**Coded Symbol**	**Constraint**		
Essential oil volume (mL)	Y	Maximize		

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
