# Peer review of "Selection of Optimal Operating Conditions for Extraction of Myrtus Communis L. Essential Oil by the Steam Distillation Method"

_molecules, 2020, doi:10.3390/molecules25102399_

Round 1

Reviewer 1 Report

Authors corrected their manuscript carefully and now, it is suitable for publication in Molecules.

Reviewer 2 Report

Authors made all corrections suggested by reviewers and now the paper is improved. For this reason I suggest to publish the paper as it is in the last form.

This manuscript is a resubmission of an earlier submission. The following is a list of the peer review reports and author responses from that submission.

Round 1

Reviewer 1 Report

Title: Selection of Optimal Operating Conditions for Extraction of Myrtus  Communis L. Essential Oil by Steam Distillation Method

In my opinion this paper is very technical study that deals with Food Engineering, that is not a topic of Molecules journal. Authors made an optimization of steam distillation method without a comparison with other techniques used in literature. In this application there is no novelty at all because more Authors report optimization of essential oil from Myrtle.

Abstract: Following the optimum parameters, the GC-MS assay proved the high purity of the Myrtus communis L. essential oil with two predominants components, α-Pinene – 33.14% and Eucalyptol – 55.09%.

It is not demonstred that “two predominants components, α-Pinene – 33.14% and Eucalyptol – 55.09%” give the “high purity” of EO from Myrtle.

Abstract: The approach suggested in this paper could be extended for the optimization at industrial scale of a wide variety of essential oils obtained from vegetable products.

I do not agree with this conclusion because the optimization at industrial scale is not simply an expansion of system but it needs to reconsider all parameters for biggest plants.

Literature

Scopus

EXPORT DATE:14 Feb 2020

Zermane, A., Larkeche, O., Meniai, A.-H., Crampon, C., Badens, E.

35605129300;15837066800;24178130000;6507158958;55955616200;

Optimization of essential oil supercritical extraction from Algerian Myrtus communis L. leaves using response surface methodology

(2014) Journal of Supercritical Fluids, 85, pp. 89-94. Cited 24 times.

https://www.scopus.com/inward/record.uri?eid=2-s2.0-84889643114&doi=10.1016%2fj.supflu.2013.11.002&partnerID=40&md5=e45f2378fc6101eef7565ea934c27c55

DOI: 10.1016/j.supflu.2013.11.002

DOCUMENT TYPE: Article

PUBLICATION STAGE: Final

SOURCE: Scopus

Pereira, P., Bernardo-Gil, M.G., Cebola, M.J., Mauricio, E., Romano, A.

7202938207;6701619475;6603503233;55863971900;56249997600;

Supercritical fluid extracts with antioxidant and antimicrobial activities from myrtle (Myrtus communis L.) leaves. Response surface optimization

(2013) Journal of Supercritical Fluids, 83, pp. 57-64. Cited 22 times.

https://www.scopus.com/inward/record.uri?eid=2-s2.0-84884565093&doi=10.1016%2fj.supflu.2013.08.010&partnerID=40&md5=a0ac150601472885854d86e722259a4a

DOI: 10.1016/j.supflu.2013.08.010

DOCUMENT TYPE: Article

PUBLICATION STAGE: Final

SOURCE: Scopus

Ghasemi, E., Raofie, F., Najafi, N.M.

14622506900;6507337513;55662871600;

Application of response surface methodology and central composite design for the optimisation of supercritical fluid extraction of essential oils from Myrtus communis L. leaves

(2011) Food Chemistry, 126 (3), pp. 1449-1453. Cited 60 times.

https://www.scopus.com/inward/record.uri?eid=2-s2.0-78751580469&doi=10.1016%2fj.foodchem.2010.11.135&partnerID=40&md5=f7a8e0c4eab592a664ff4c9bc262be26

DOI: 10.1016/j.foodchem.2010.11.135

DOCUMENT TYPE: Article

PUBLICATION STAGE: Final

SOURCE: Scopus

Ammar, A.H., Zagrouba, F., Romdhane, M.

57198277867;6602801058;6701846394;

Optimization of operating conditions of Tunisian myrtle (Myrtus communis L.) essential oil extraction by a hydrodistillation process using a 24 complete factorial design

(2010) Flavour and Fragrance Journal, 25 (6), pp. 503-507. Cited 14 times.

https://www.scopus.com/inward/record.uri?eid=2-s2.0-77958566795&doi=10.1002%2fffj.2011&partnerID=40&md5=7d3da66c87074f4b614ee0accc6c06b8

DOI: 10.1002/ffj.2011

DOCUMENT TYPE: Article

PUBLICATION STAGE: Final

SOURCE: Scopus

Haj Ammar, A., Zagrouba, F., Romdhane, M., Abderrabba, M.

36008622800;6602801058;6701846394;6603480414;

Extraction of myrtle (Myrtus communis L.) essential oil from Tunisia by hydrodistillation [Extraction de l'huile essentielle de myrte (Myrtus communis L.) provenant de la Tunisie par hydrodistillation]

(2010) Acta Horticulturae, 853, pp. 241-250.

https://www.scopus.com/inward/record.uri?eid=2-s2.0-77951834417&partnerID=40&md5=f5376d8ec045a14f74867770b839d398

DOCUMENT TYPE: Article

PUBLICATION STAGE: Final

SOURCE: Scopus

However the method applied from Authors is not new and for this application results are not valided in respect of a different solid-liquid extraction technique like carbon dioxide supercritical extraction,in terms of yield of extraction and maceration in ethyl alchol for comparison of substances extracted from Myrtle.

Research approach is generic and not scientific because Authors find the optimal conditions for the obtaining the best recovery in terms of volumes of essential oil extracted but do not consider the “quality” of essential oil in terms of distribution of terpenes.

In my opinion the paper has serious flaws in experimental part and needs to be improved for the pubblication in Molecules.

For these reasons I suggest to not pubblish paper in this form.

Reviewer 2 Report

Novelty of the investigation is rather poor . The Authors took known and well described plant, used commercially available equipment and statistical  method to optimize extraction condition. Taguchi’s approache and RSM are not new methods and they are commonly used for optimization purpose. Moreover, only one factor (volume of oil) was taking into consideration to establish “the optimal conditions”.  Meanwhile, the quality (chemical composition)  of essential oil is also important factor.  The paper needs a lot of improvement and the investigation should be supplemented.

The detailed suggestions:

-the novelty of the study should be more highlighted (see comment above)

There are a lot of not precise and confusing statements in manuscript:

Abstract: “

- „Thus, this study targeted the selection of optimal operating  conditions in order to obtain high volumes of essential oils using a Box-Wilson experimental plan” –“a Box-Wilson experimental plan” was used to select the optimal condition (not to obtain essential oil, because the oil was obtained using steam distillation).

- (…) to improve the quality of the extraction process of essential  oil (…)” – it should be: to improve the extraction efficacy”

- the last sentence in Abstract is unnecessary because it is only Authors’ speculation not supported by data.

Introduction:

Line 44: the mentioned compounds are not components of essential oil Line 58-69 are unnecessary in this place because they are not directly associated with the aim of the work. They could be used as a part of discussion to compare the obtained results with literature. Line 81: RSM is not a method to obtain the essential oil – reedit the sentence. Line 82 - 101: Short mention on application of RSM to optimize essential oil extraction with some examples is ok but there is no need to give details of procedures obtained by other researches (temperature, pressure, time),  because they used quite different plant material and probably different extraction technique. Instead some information on techniques used to obtain Myrtus essential oil should be added. Line 104: “to the best of our knowledge” - should be moved to the beginning of the sentence The last sentence of Introduction: I understand that Authors want to show that they used previously RSM and Taguchi approaches but I think that it is mentioned in wrong place. Move the sentence to e.g. line 104 and use rather the expression “ this approach” or “this method” (instead of “technique”).

Results and discussion.

-          The discussion is poor. First of all, lack of comparison with literature data, e.g. in term of comparison the extraction efficacy using the proposed procedure with the results obtained in the other investigations. The discussion should be expanded.

-          Lines 113-129 (and Table 1) are description of methodology (not results) and should be place in Methods section.

-          The comparison of chemical composition of essential oil obtained using different extraction conditions would be the most interesting and valuable part of the study.

Methods:

-          The detailed description of equipment is unnecessary – the data of manufacturer and the type of apparatus is enough.